# Macrophage-Secreted CSF1 Transmits a Calorie Restriction-Induced Self-Renewal Signal to Mammary Epithelial Stem Cells

**DOI:** 10.3390/cells11182923

**Published:** 2022-09-19

**Authors:** Anna Kosenko, Tomer Meir Salame, Gilgi Friedlander, Itamar Barash

**Affiliations:** 1The Volcani Center, Agricultural Research Organization, Institute of Animal Science, Bet Dagan 50250, Israel; 2The Robert H. Smith Faculty of Agriculture, Food and Environment, The Hebrew University of Jerusalem, Jerusalem 9190501, Israel; 3Life Sciences Core Facilities, Weizmann Institute of Science, Rehovot 7632706, Israel; 4The Mantoux Bioinformatics Institute of the Nancy and Stephen Grand Israel National Center for Personalized Medicine, Weizmann Institute of Science, Rehovot 7632706, Israel

**Keywords:** mammary gland, stem cell, calorie restriction, macrophage, monocyte, CSF1, RNA sequencing, niche

## Abstract

Calorie restriction enhances stem cell self-renewal in various tissues, including the mammary gland. We hypothesized that similar to their intestinal counterparts, mammary epithelial stem cells are insulated from sensing changes in energy supply, depending instead on niche signaling. The latter was investigated by subjecting cultures of mammary epithelial stem cells for 8 days to in vivo paracrine calorie-restriction signals collected from a 4-day-conditioned medium of individual mammary cell populations. Conditioned medium from calorie-restricted non-epithelial cells induced latent cell propagation and mammosphere formation—established markers of stem cell self-renewal. Combined RNA-Seq, immunohistochemistry and immunofluorescence analyses of the non-epithelial population identified macrophages and secreted CSF1 as the energy sensor and paracrine signal, respectively. Calorie restriction-induced pStat6 expression in macrophages suggested that skewing to the M2 phenotype contributes to the sensing mechanism. Enhancing CSF1 signaling with recombinant protein and interrupting the interaction with its highly expressed receptor in the epithelial stem cells by neutralizing antibodies were both affected stem cell self-renewal. In conclusion, combined in vivo, in vitro and in silico studies identified macrophages and secreted CSF1 as the energy sensor and paracrine transmitter, respectively, of the calorie restriction-induced effect on mammary stem cell self-renewal.

## 1. Introduction

A reservoir of bipotent and lineage-restricted mammary epithelial stem cells (MaSCs) drives morphogenesis and homeostasis of the mammary ductal tree, respectively [1,2]. Nevertheless, individual MaSCs are not distinguishable in the epithelial cell population and numerous studies have been aimed at enriching this population of cells (reviewed in [3]). Consequently, commonly used marker pairs CD24 and CD49f or CD44 and CD49f have been shown to enrich the MaSC population, which can repopulate de-epithelialized mammary stroma [4,5]. In vitro analyses of mammospheres generated under non-adherent conditions and cell-propagation assays have been established as reliable tools to monitor human [6], mouse [7,8] and bovine [9,10] MaSC self-renewal. 

Mammary-specific regulation of epithelial stem cell self-renewal by reproductive steroid hormones has been identified [11]. In addition, the activity of common self-renewal signals that affect stem cells in other tissues has also been demonstrated. Calorie restriction (CR), i.e., the limitation of energy (food) intake without malnutrition, has been extensively studied in the context of extending human and animal life spans and slowing the rate of aging ([12,13,14] and references therein). This beneficial aspect of a controlled decrease in energy supply involves induced self-renewal of undifferentiated somatic stem cells [15,16] that are able to replenish apoptotic tissue cells during homeostasis, repair damaged tissues [17] and inhibit the development of aging diseases [18,19]. CR has also been recently reported as a non-pharmacological intervention in induced and spontaneous cancers [20]. In a mouse model, it inhibited the growth of induced and spontaneous tumors, including breast cancer [21]. The mechanisms mediating this beneficial effect are not clear; it may involve a unique immune signature of CR and altered activation of metabolic pathways, such as AMPK/mTOR [20,22,23]. Nevertheless, direct association of the anti-tumorigenic effect of CR with stem cell activity has not been demonstrated. 

Indirect sensing of the decrease in energy supply by stem cells has been clearly demonstrated in the intestine, where individual stem cells at the bottom of the crypt are distinguished by the Lgr5 marker. Mechanistically, the reduced calorie intake causes decreased activity of the energy sensor mTOR in neighboring Paneth epithelial cells, which is translated into cADP ribose paracrine signaling for self-renewal of their juxtaposed stem cell ancestors [24,25,26]. CR has also been shown to induce MaSC self-renewal in mice and in bovine parenchymal tissue implanted in de-epithelialized mouse mammary gland [10].

The leading hypothesis of this study is that, similar to their intestinal counterparts, MaSCs are insulated from sensing changes in the energy supply to the tissue, and, in fact, depend on signals from a specific niche cell population. The intestinal energy-sensing and signaling Paneth cells are tissue-specific. Therefore, we aimed here to identify a corresponding energy-sensing niche and associated signaling to the MaSCs. 

MaSCs reside in the basal region of the ducts, possibly between the basal and luminal layers [1,2,27]. Hence, these cells are subjected to signals from several candidate cell populations: differentiated luminal cells and their progenitors, basal cells and non-epithelial cells. For example, steroid hormones were found to induce an 11-fold increase in MaSC number during pregnancy through paracrine signaling by the RANK ligand secreted by mature luminal cells [15]. In addition, Joshi et al. [28] demonstrated that progesterone propels MaSC expansion during the estrous cycle through paracrine signaling by the RANK ligand and WNT4, which are also secreted by luminal cells. Recently, the myoepithelium-secreted protease, Adamts18, has been reported to link steroid hormone action to MaSC niche activation [29]. Signals from non-epithelial cells also seem to be important for MaSC maintenance and expansion. Macrophages have been reported to be components of the spermatogonial and hematopoietic stem cell niches [30,31]. These monocyte-derived cells of the myeloid lineage play an important role in the innate immune defense system, but also have roles in tissue remodeling during development and in normal tissue homeostasis [32,33]. In the mammary gland, macrophages are required for branching morphogenesis ([34] and reviewed in [35]) and are essential for MaSC self-renewal [36]. CSF1 has been proposed to serve as a chemoattractant for macrophages to the stem cell niche. Recently, direct functional cross talk has also been reported between MaSCs and their macrophage niche through Dll1/Notch-mediated signaling [37,38].

Here, we report the identification of a mammary macrophage population that senses CR, polarizes to the M2 phenotype and transmits a self-renewal signal through CSF1 secretion to the MaSCs.

## 2. Materials and Methods

### 2.1. Mice and CR

Female virgin FVB/N mice (6 weeks old; ~21 g) were used in this study. They were housed under a 12 h light/dark cycle and fed ad libitum. Mice were subjected to CR for 8 weeks by receiving 70% of the food consumed by their ad libitum-fed counterparts on the previous day, with vitamin and mineral (Pfizer, Aprilia, Italy) supplementation. In each experiment, 20 mice were allocated to the CR group and a similar number of 20 mice served as their respective ad libitum-fed controls. MaSCs were separately extracted and collected from 32 mice. All animals used in this study were treated humanely. Study protocols were in compliance with the regulations of the Israeli Ministry of Health and local institutional policies (approval code 793/18). 

### 2.2. Flow Cytometry

Mouse #4 mammary glands were harvested and digested, as previously described [7,10]. Lineage^−^ (Lin^−^) cell suspension was prepared using the EasySep mouse epithelial enrichment kit according to the manufacturer’s protocol (StemCell Technologies, Vancouver, CB, Canada). Through magnetic separation, upon which the Lin^-^ procedure is based, endothelial and some hematopoietic and fibroblast cells were eliminated before sorting. Dispersed cells (10^7^ cells/mL) were resuspended in Hank’s Balanced Salt Solution (HBSS) containing 5% fetal bovine serum (FBS) (Biological Industries, Beit Haemek, Israel); they were labeled with phycoerythrin (PE)-conjugated anti-CD24 and fluorescein isothiocyanate (FITC)-conjugated anti-CD49f antibodies to identify MaSCs, as well as other epithelial and non-epithelial cell populations, or with PE-conjugated anti-CD24 and FITC-conjugated anti-CD14 antibodies to enrich for monocytes/macrophages. The antibodies, their sources and dilutions are given in Appendix A. Fixable viability stain (FVS450; BD Biosciences, Bedford, MA, USA; diluted 1:500) was used to mark dead cells, which were eliminated from the cell separation. Cell clumps were excluded by filtration through a 40 μm mesh (Falcon, Bedford, MA, USA). Cell sorting was performed in a FACS Aria III cell sorter (BD Biosciences) at the Department of Biological Services of the Weizmann Institute of Science (Rehovot, Israel), as previously described [7,10,39]. Resulting data were visualized and analyzed by FACSDiva and FlowJo software (BD Biosciences). 

### 2.3. Cell Culture

To determine the carryover effects of the in vivo CR experimental period on MaSC properties, these cells were subjected to the conditioned media of the individually sorted mammary cell populations (i.e., luminal cells, luminal progenitor cells, basal cells and non-epithelial cells). Conditioned media were collected from 4-day cultures (unless otherwise indicated) of mammary epithelial or non-epithelial cells, which were derived from the #4 mammary glands of CR mice and their ad libitum-fed counterparts (see Section 3.1 for experimental details). The conditioned media contained a collection of paracrine-secretion factors that distinguished the CR-affected glands from their controls and were tested for long-term effects on stem cell self-renewal by analyzing propagation rate and mammosphere development under non-adherent conditions. To harvest the conditioned medium, cells from each of the sorted populations were seeded in a 96-well cell-culture plate (Corning, Lowell, MA, USA) at the concentrations indicated in the legends to the relevant figures. The culture medium (mammary medium) consisted of DMEM-F12 containing 5% FBS, hydrocortisone (0.5 μg/mL, Sigma, St. Louis, MO, USA), insulin (5 μg/mL, Sigma), gentamicin (50 μg/mL, Biological Industries), streptomycin and penicillin (100 μg/mL and 100 U/mL, respectively, Biological Industries), human epidermal growth factor (10 ng/mL, Merck, Darmstadt, Germany), human fibroblast growth factor 2 (10 ng/mL, Merck), heparin (4 μg/mL, Merck), cholera toxin (10 ng/mL, Sigma) and human leukocyte antigen B27 (4 μL stock/mL, Invitrogen, Carlsbad, CA, USA) [9]. Upon termination of the 4-day culture, the conditioned medium from each cell population was collected, centrifuged to eliminate cell debris and frozen at −80 °C. 

For propagation rate analysis, freshly sorted MaSCs were seeded in a 96-well cell-culture plate (Corning) at a density of 350–500 cells/well and cultured for 8 days (unless otherwise indicated) in conditioned medium harvested from cultures of the various cell populations. Cells were trypsinized at the indicated times, counted and, where noted, re-seeded at the indicated density for further follow-up. For the mammosphere assay, the sorted MaSCs were seeded in a 96-well ultra-low-attachment plate (Corning) at a density of 350–500 cells/well and supplemented with conditioned mammary medium. Developing mammospheres were visualized with an Eclipse Ti inverted fluorescence microscope (Nikon Instruments, Melville, NY, USA), equipped with NIS-Elements AR 3.2 imaging software.

### 2.4. Immunohistochemistry and Immunofluorescence 

Mouse #4 mammary glands were excised and fixed in Bouin’s solution. Fixed tissues were dehydrated in a graded ethanol series (50% to 100%), cleared in xylene and embedded in paraffin. Immunohistochemistry was performed on 5 μm thick paraffin-embedded sections after antigen retrieval. The sections were first blocked with the Fab fragment of goat anti-mouse IgG diluted 1:20 (Jackson ImmunoResearch Labs, West Grove, PA, USA) and then incubated overnight with the primary antibody at 4 °C. The antibodies and their dilutions are given in Appendix A, and their ligands are described in the relevant experiment. The paraffin sections were then washed and incubated with N-Histofine (Nichirei Biosciences, Tokyo, Japan) for 30 min at room temperature. Signals were generated with 3,3′-diaminobenzidine (DAB) substrate (Vector Laboratories, Burlingame, CA, USA). Nuclei were stained with hematoxylin (Sigma). For immunofluorescence analyses of fixed tissues, paraffin sections were blocked and incubated with the primary antibody, as described above, and then incubated with secondary antibodies for 1 h at room temperature. Nuclei were stained with 4′,6-diamidino-2-phenylindole (DAPI) (Qbiogene, Irvine, CA, USA). For immunofluorescence analyses of cultured cells, mouse mammary cells were fixed with 4% paraformaldehyde, washed with PBS and treated with 0.5% Triton X-100 (BDH, Poole, England) for 5 min. Following overnight incubation in blocking solution (2% goat serum and 1% bovine serum albumin, *v*/*v*, in PBS) at 4 °C, the fixed cultures were reacted with primary antibodies for 1 h at room temperature and then overnight at 4 °C. Incubation with secondary antibodies (Appendix A) proceeded for 1 h at room temperature, and nuclei were stained with DAPI. In all of the analyses, stained tissues were visualized and photographed under the Eclipse Ti inverted fluorescence microscope or an Olympus IX 81 inverted laser scanning confocal microscope (FluoView 500, Tokyo, Japan).

### 2.5. RNA Extraction, Sequencing and Bioinformatics 

Poly(A) RNA was extracted from 15,000 non-epithelial cells of an individual mouse #4 mammary gland using the Dynabeads mRNA DIRECT purification kit (Thermo Fisher, Waltham, MA, USA) according to the manufacturer’s protocol. Briefly, Lin^-^CD49f^neg^CD24^neg^ cells were sorted into individual tubes containing 40 μL lysis buffer. The poly(A) RNA was captured on poly-dT-bound magnetic beads. The beads were extensively washed and the bound RNA was eluted with 6.5 μL of 10 mM Tris-HCl (pH 7.5) buffer heated to 85 °C for 2 min. 

A bulk adaptation of the MARS-Seq protocol [40,41] was used to generate RNA-Seq libraries for expression profiling at the Crown Genomics Institute (Nancy and Stephen Grand Israel National Center for Personalized Medicine, Weizmann Institute of Science). Briefly, 30 ng of input RNA from each sample was barcoded during reverse transcription and pooled. Following Agencourt AMPure XP bead cleanup (Beckman Coulter, Indianapolis, IN, USA), the pooled samples underwent second-strand synthesis and were linearly amplified by in vitro T7 transcription. The resulting RNA was fragmented and converted into a sequencing-ready library by tagging the samples with Illumina sequences during ligation, reverse transcription and PCR. Libraries were quantified by Qubit and TapeStation, as well as by quantitative PCR for the housekeeping gene *ActB*. Single-end reads were sequenced in an Illumina NextSeq machine. Poly-A/T stretches and Illumina adapters were trimmed from the reads using cutadapt [42] and resulting sequences shorter than 30 bp were discarded. Remaining reads were mapped to the mm10 genome using STAR [43] with EndToEnd option and outFilterMismatchNoverLmax set to 0.05. De-duplication was carried out by flagging all reads that were mapped to the same gene and had the same unique molecular identifier (UMI). Reads were counted using Refseq annotations (1000 bases of the 3′ UTR) with htseq-count [44] and corrected for UMI saturation. Differentially expressed genes were identified using DESeq2 [45] with the betaPrior, cooksCutoff and independentFiltering parameters set to False. Raw *p* values were adjusted for multiple testing using the procedure of Benjamini and Hochberg. Pipeline was run using snakemake [46].

Heatmaps were produced as follows. For each gene, the DESeq2 normalized counts (log2 scale) were standardized to have zero mean and one-unit standard deviation. K-means partitioning clustering was performed with Euclidean distance measure. The expression profile is accompanied by a colored bar, indicating the standardized log2 normalized counts. GO analysis enrichment was performed with Mousemine (Mouse Genome Informatics, The Jackson Laboratory, Bar Harbor, Maine (http://www.mousemine.org (accessed on 22 April 2022)).

### 2.6. Statistical Analysis

Unless otherwise indicated, t-test was performed for statistical analyses.

### 2.7. Accession Numbers 

The RNA-Seq data discussed in this publication have been deposited in NCBI’s Gene Expression Omnibus [47] and are accessible through GEO Series accession number GSE207481.

## 3. Results

### 3.1. The Inductive Effect of CR on MaSC Self-Renewal Is Mediated through Paracrine Signaling by the Non-Epithelial Cell Population

The hypothesis of the current study envisages a mammary cell population that senses the rate of energy flow to the tissue and mirrors CR in paracrine fashion to incompetent neighboring stem cells to promote self-renewal. To identify this hypothesized population and the involved signaling, evidence for the promotion of MaSC self-renewal was sought in the conditioned medium of candidate niche cell populations, seeded immediately after the in vivo CR experimental period (Figure 1A). Thus, female mice were subjected to CR by limiting their food consumption to 70% of their ad libitum-fed counterparts for 8 weeks. Following this period, their weight was 30% lower than that of their respective controls (Figure 1B). Mice were then sacrificed and #4 mammary glands were excised. Single-cell suspension was prepared and Lin^-^ mammary cell populations, devoid of endothelial cells and fibroblasts, as well as red blood cells and most fat cells, which were eliminated during the preparation and were subjected to flow cytometry using the CD24-labeled and CD49f-labeled antibodies (Figure 1C). In addition to the CD24^med^CD49f^high^ MaSCs, four candidate cell populations, compatible with mediation of stem cell self-renewal, were sorted: CD24^low^CD49f^high^ basal cells, CD24^high^CD49f^med^ luminal progenitors, CD24^med^CD49f^med^ differentiated luminal cells and CD24^neg^CD49f^neg-low^ non-epithelial cells [4,5,7,48,49,50]. Of these, the candidate cell population was determined by following the latent effects of its conditioned medium, collected after a 4-day incubation period, on cell propagation and mammosphere formation in cultures initiated from MaSCs. Importantly, the conditioned medium was only collected during the first days of culture to associate its long-term effects on the extracted stem cells and not on their subsequently differentiated progeny. As demonstrated in Figure 1D,E, MaSC self-renewal parameters (i.e., cell propagation and mammosphere generation) were significantly induced (15- and ~2-fold, respectively) only after the addition of conditioned medium collected from the non-epithelial cell population. 

The involvement of the non-epithelial cell population in mediating the inductive effect of CR on MaSC self-renewal was further characterized by applying longer incubation periods. Figure 2A illustrates the experimental design. Conditioned medium was collected from 4-day cultures of ad libitum-fed (control) and CR non-epithelial mammary cells to analyze the effect of continually secreted materials from the in vivo CR period on stem cell properties. To rule out the intrinsic effects of the culture and confirm carryover of the in vivo effect, conditioned medium was also collected on days 4–5, 4–6 and 4–7 of the culture. It was then individually supplemented to the cultures of the MaSC population for the first 8 days of culture for time-course monitoring of the latent effect of the in vivo CR period on cell propagation. Ten days after seeding, an initial induction in cell proliferation characterized both CR- and control-derived cultures (Figure 2B). This most likely represents an induced short-term proliferation of stem cells or both stem cells and their partly differentiated co-purified progenitors. During longer incubation, the initial inductive effect of CR on long-term stem cell self-renewal continued, the partly differentiated progenitor effect was gradually lost and the supplementation of conditioned medium derived from the non-epithelial CR cells resulted in a latent improvement of cell propagation and culture maintenance, compared to its respective control. This indicated an initially higher number of MaSCs in vivo. Extending the CR procedure in culture by diluting the CR conditioned medium to 65% did not affect propagation rate at the end of the culture period. In contrast to the significant effect of conditioned medium collected for the first 4 days of culture, no effects were observed for the supplementation of non-epithelial CR conditioned medium collected after the 4th day of culture (Figure 2C), or on days 4–6 (Figure 2D) or 4–7 (Figure 2E) of culture. This finding supports limited carryover of the in vivo CR effect to the initial 4-day culture period. The number of mammospheres generated from sorted stem cells, cultured in conditioned medium of non-epithelial cells, derived from CR mammary glands, was also significantly higher than that from their respective control. No significant difference was observed between the number of mammospheres developed in diluted, compared to non-diluted, medium (Figure 2F). In addition, no significant difference was observed between the number of mammospheres developed in conditioned medium of ad libitum-fed mice and fresh medium (not shown). 

### 3.2. CR Induces Csf1 Expression in Non-Epithelial Mammary Cells

Next, gene expression analysis was performed to identify a putative CR-responsive gene in the non-epithelial cell population that encodes a secreted protein with a positive effect on MaSC self-renewal. Non-epithelial mammary cells were sorted from dispersed cells of #4 mammary glands of individual CR and ad libitum-fed control mice. Poly(A) RNA was extracted and subjected to RNA-Seq analysis. Hierarchical clustering distinguished the two groups (Figure 3). Overall, 1067 genes were differentially expressed between non-epithelial mammary cells from CR and ad libitum-fed control mice at adjusted *p* ≤ 0.05 (Figure 3A and Appendix A and Table 1). More genes (581 genes, representing 54% of the total number) were identified with relative higher expression level in cells derived from ad libitum-fed mice, compared to their CR counterparts, whereas 486 genes (46% of the total number) were relatively highly expressed in the CR cells. By comparison to a combined dataset of genes encoding secreted proteins (UniPort, the universal knowledge base [51], Ingenuity Pathway Analysis [IPA; Qiagen, Redwood City, CA, USA] https://digitalinsights.qiagen.com/IPA (accessed on 26 April 2022) and the Human Genome Atlas), 151 differentially expressed genes encoding secreted proteins were identified (Figure 3B and Appendix A, tab2). In contrast to the minor negative effect of CR on the total number of differentially expressed genes, only 48 of the genes encoding secreted proteins (32% of the total number) were expressed at relative higher level in the ad libitum-derived non-epithelial cells, compared to their CR counterparts, whereas 103 genes (68% of the total number) were relatively highly expressed in cells derived from CR mice. 

Gene ontology (GO) enrichment analysis identified compatibility between the main biological functions, characterizing the total number of differentially expressed genes (Figure 3A) and those of the encoded secreted proteins (Figure 3B). Genes that were relatively highly expressed in non-epithelial mammary cells of the ad libitum-fed mice were mainly involved in immunological aspects: T cell, lymphocyte and leukocyte activity. CR dampened this immune specificity by decreasing the expression of genes involved in immunological processes. For example, a 12- to 14-fold decrease was observed in the expression of *Cd6, Lta, Def6, Ccl5 and Cd69*. Consequently, an alternative variety of cellular activities emerged as significant in cells derived from the CR mice, ranging from cell motility and locomotion to epithelial development and tube formation. This CR-dependent shift in the enriched activities was confirmed by IPA of physiological system development and function (Appendix A) that also identified major regulators of this process.

To identify distinct candidate genes, encoding paracrine secreted proteins, that induce stem cell self-renewal, the tool “My pathway” in IPA was applied. It and related genes encoding secreted proteins that are induced by CR to three lists of stem cell markers: (i) mouse mammary stem cell markers [7], (ii) combined mouse and human mammary stem cell markers [7] and (iii) stem cell markers from various mouse tissues [52]. As demonstrated in Table 1, all three interactions between the CR-induced genes and the stem cell markers identified *Csf1* as a potential regulator of stem cell self-renewal activity. Its candidacy was further supported by its highly significant 2-fold higher expression in the CR-induced non-epithelial cells, as compared to ad libitum-fed controls (Figure 4A). Lower expression levels of *Csf1* in the MaSCs themselves, as compared to their differentiated counterparts (Figure 4A, 2.2–8 fold) was calculated from a previously generated dataset [7] that distinguishes MaSCs from their immediate progenitors and from more differentiated cells according to the expression of CD24, CD49f, CD200 and CD200R. Inversely, the expression of the Csf1 receptor (*Csf1R*) was almost undetected in the non-epithelial mammary cells, regardless of CR (Figure 4B). Importantly, *Csf1R* expression was detectable and significantly higher in MaSCs, as compared to their immediate progenitors and the differentiated epithelial cells (Figure 4B and [7]). These data combined current and previous analyses to identify *Csf1* expression in non-epithelial mammary cells, as a potential mediator of CR’s inductive effect on MaSC self-renewal.

### 3.3. CR-Induced CSF1 Expression Is Detected in the Mammary M2 Macrophage Population

The Lin^-^ non-epithelial CD24^neg^CD49f^neg-low^ mammary cells represent a heterologous population, a considerable fraction of which is associated with cell immunity. To identify immune cells involved in the CR-induced CSF1 secretion, this cell population was further dissected and CD24^neg^ cells were sorted and collected according to the expression of CD14—an established marker of the myeloid lineage-derived monocytes/macrophages [53,54,55]. A CD24^neg^CD14^high^ (P1) cell subpopulation was sorted (Figure 5A), cultured, fixed and confirmed by immunofluorescence analysis to contain cells expressing CD14, but not CD3 or CD19 (Figure 5B,B′). CD3 and CD19 are established markers of the lymphoid lineage-derived T cells [56,57] and B cells [58,59], respectively. The cultured CD24^neg^CD14^low^ (P2) population did not express detectible levels of CD14 and was composed of cells expressing high levels of CD3. CD19-expressing B cells were detected only in CD24^high^ cell populations (P4) that represent luminal progenitors [60]. The P3 population consisted of differentiated luminal epithelial cells. 

Figure 5E demonstrates the proportions of these cell subpopulations among the sorted single live Lin^–^ mammary cells. The CD24^pos^ and CD24^neg^ cells represent comparable halves of the sorted populations in the mammary glands of both ad libitum-fed and CR mice. Within the non-epithelial CD24^neg^ population, CR induced the number of CD14^pos^ (P1) myeloid monocytes/macrophages by 65%. Conversely, the CD14^low-neg^ lymphoid cell (P2) population was larger in the ad libitum-fed control mammary gland by a comparable proportion (69%). Of note, there were no significant differences (*p* < 0.05) between the proportions of the subpopulations derived from the ad libitum-fed and CR mice in any of these comparisons. 

In vivo localization of the defined CD24^neg^ subpopulations was also studied. CD14^pos^ monocytes/macrophages were identified by immunohistochemical analysis around and between the luminal and basal layers of the ducts, as well as within the lymph node (Appendix A). Immunofluorescence analysis demonstrated induced CSF1 expression in the mammary glands of CR mice (Figure 6A′,A″) and in 3-day cultures of the P1 (monocyte and macrophage) population derived from CR mice, as compared to their ad libitum-fed counterparts (Figure 6B,B′). CSF1R was highly expressed in the cultured MaSCs but was not detected in the cultured monocyte/macrophage population (Figure 6C,C′). 

The F4/80 glycoprotein marks differentiated macrophages [34]. It was co-expressed in most CD14^pos^ cells and also detected by immunofluorescence analysis in the epithelial mammary ducts and lymph node (Figure 7A,B and Appendix A). The few differentially stained cells may represent completely undifferentiated myeloid cells that do not express F4/80, or fully differentiated cells in which the F4/80 was not detected for technical reasons. M1/M2 macrophage polarization toward pro-inflammatory or anti-inflammatory (wound-healing) states, respectively, plays an important role in regulating tissue homeostasis. The pStat1 and pStat6 expression identifies the respective phenotypes [61,62,63]. CR glands maintained a high number of strongly pStat6-stained macrophages/monocytes, as compared to the scarce and low pStat6-stained monocytes/macrophages in the ad libitum-fed mice (Figure 7B,B′), suggesting dominance of the M2 phenotype upon CR. The pStat1 expression was undetected in the virgin gland (Figure 7C). Its expression in the pregnant gland served as a positive control in this analysis (Figure 7C′). 

### 3.4. Manipulating CSF1 and CSF1R Activities Affects MaSC Self-Renewal

CR-induced CSF1 expression was detected in the non-epithelial monocyte/macrophage cell population in vivo and in the subsequent culture, thus confirming the RNA-Seq analysis. In contrast, CSF1R was detected in MaSCs, but not in monocytes/macrophages. The involvement of CSF1, acting via its cognate receptor, in MaSC self-renewal was further studied by the complementary analyses of cell propagation and mammosphere generation under non-adherent conditions (Figure 8). Here, CSF1 activity was induced by the addition of recombinant CSF1 to cultures of freshly isolated MaSCs, while CSF1R expression was downregulated in the highly expressing MaSCs by neutralizing antibodies [64]. CSF1 supplementation for the first 8 days to cultures stemming from MaSCs induced latent cell propagation (Figure 8A) and mammosphere formation (Figure 8B) in a dose-dependent manner. Differences of up to 1.6- or 1.7-fold, respectively, were still measured at the end of the respective 70- and 54-day culture periods. Conversely, addition of CSF1R neutralizing antibodies to MaSC cultures resulted in a comparable decrease in both cell propagation rate and mammosphere formation by 1.6- and 1.8-fold, respectively, by the end of the culture period (Figure 8C,D).

## 4. Discussion

MaSCs are metabolically crippled, compared to their common progenitors [7] and depend on their differentiated luminal and basal descendants to receive self-renewing renewal steroid hormone signals [11,15,28,29]. CR induces self-renewal of MaSCs [10], but no evidence was available for a comparable absence of MaSC sensor that detects changes in the energy status of the tissue or for a putative niche that transmits specific self-renewal signals in response to the CR. Nevertheless, previous observation indicated that intestinal Paneth cells do transmit a CR signal for self-renewal of their neighboring Lgr5^+^-defined stem cells [24,25]. Seeking a comparable mammary detection and signaling system, all of the CR paracrine signals were collected here in conditioned medium of short-term cultures from individual mammary cell populations derived from CR mice and their ad libitum-fed counterparts. A significant self-renewal-inducing effect of CR was only exerted by conditioned medium from the non-epithelial cell population. In contrast to the differentiated mammary epithelial cells that transmit steroid hormone signals, or the intestinal signaling Paneth cells, this population has no lineage association with their stem cell ancestors and its contribution to mammary gland development ranges from passive support to trophic factor supplementation. RNA-Seq analysis of this hypothetical niche was performed to identify a gene that codes for a secreted protein, which meets the characteristics of a paracrine mediator of CR-induced stem cell self-renewal. Concurrent with the substantial content of hematopoietic cells in the analyzed non-epithelial cell population and their reported repressive effect on immune functions [65,66,67], CR decreased the expression of genes that mediate immunity, and especially those encoding secreted proteins, to a level that did not allow the identification of this unique activity in the cell metabolism. On this background, significant 2-fold induction was detected in the expression of a gene encoding the secreted protein CSF1, which was associated with induced MaSC self-renewal markers. CSF1 is a hematopoietic growth factor that binds the tyrosine kinase receptor *c-fms* and induces myeloid macrophage proliferation [68,69]. Nevertheless, it has also been reported [70] that exposure of monocytes (macrophage ancestors), but not T-lymphocytes, to γ interferon or phorbol myristate acetate induces endogenous CSF1 expression in the monocytes themselves, suggesting that these cells, similar to T cells [71] and B cells [72], can be induced to release growth factor, specifically for their own lineage. Accordingly, by combined RNA-Seq, flow cytometry, immunohistochemical and immunofluorescence analyses, the inductive effect of the non-epithelial cell population on MaSC self-renewal was narrowed down to the CSF1-secreting, CD14/F4/80-expressing macrophages. Macrophages are pleiotropic cells that assume a variety of functions, depending on their tissue of residence and that tissue’s state. In addition to their phagocytic properties, they maintain homeostasis and coordinate responses to stresses, such as infection and metabolic challenge [32,73]. It has been suggested that macrophages play a role in mammary tissue remodeling by engulfing apoptotic epithelial cells during ductal morphogenesis [74,75].

The proportion of macrophages/monocytes among the non-epithelial cell population was non-significantly increased by CR, but this may only partly explain the significant 2-fold induction in CSF1 expression detected by the RNA-Seq analysis. Moreover, macrophage cultures from CR females demonstrated high CSF1 expression, compared to the almost undetected CSF1 levels in their control counterparts. Thus, CR mainly results in the induction of CSF1 expression and secretion in macrophages, contributing in a paracrine fashion to MaSC self-renewal. Interfering with the endogenous ligand–receptor interaction of CSF1–CSF1R by CSF1R-neutralizing antibodies or contributing to the CSF1 signal by the addition of recombinant CSF1 resulted in a respective decrease or increase in these MaSC self-renewal markers. Combined with the previous observation that MaSCs maintain high expression of CSF1R [7], these findings provide a mechanistic aspect to the previously published hypothesis that MaSCs require macrophage-derived factors to be fully functional [36]. Moreover, the higher activity of Stat6, the ultimate marker of M2 macrophages, which was strongly detected in mammary monocytes/macrophages of the CR mice, compared to the scarce number of pStat6-expressing cells in the respective controls, suggests skewing of this population from the M1 pro-inflammatory state to the M2 anti-inflammatory phenotype. The latter macrophage type benefits from a more efficient oxidative phosphorylation energy source that could partly compensate for reduced energy flow, as compared to the glycolysis utilized by the Krebs cycle-defective M1 phenotype (Figure 9). It has been shown that depending on the stimuli received by the microenvironment, macrophages can switch from the M1 to M2 state and vice versa [76], and that adipose tissue macrophages in lean animals activate the anti-inflammatory (M2) phenotype, whereas adipose tissue macrophages of obese mice maintain the classical pro-inflammatory (M1) type [77,78]. 

While the M1–M2 skew presents an attractive machinery for the endogenous response of tissue macrophages to CR, the associated CSF1 secretion from these cells represents the anticipated paracrine signaling, linking the macrophage energy sensor to the induction of MaSC self-renewal. Indeed, CSF1 was highly expressed in CR macrophages and induced the expression of MaSC markers. CSF1R was highly expressed in MaSCs and the effect of the CSF1–CSF1R interaction on MaSC self-renewal was modulated by external CSF1 supplementation or receptor inhibition. As the absence of CSF1 expression retards mammary tumor progression and metastasis involving mammary progenitors, but not tumor initiation that results from stem cell activity [79], there may be an additional, separate and distinct effect of CSF1 on mammary gland progenitors and the differentiated mammary epithelial cells.

A Notch–DLL1 interaction has been reported to associate macrophages with MaSCs, and WNT secretion from the macrophages induces stem cell proliferation [37,38]. The current study presents an additional route for the macrophage–stem cell interaction for the specific delivery of CR-induced self-renewal signals. The involvement of mTOR-mediated CR signaling, such as that demonstrated in the intestine [24] cannot be entirely ruled out, since expression of *Bst1*, which triggered intestinal Paneth cells’ cADP ribose paracrine secretion toward stem cell self-renewal, was induced in tissue extracts of CR-treated mammary gland (but not in rapamycin-treated ones [10]). Nevertheless, *Bst1* expression in the non-epithelial cell populations was low and unaffected. Thus, *Bst1*-mediated signaling might be exerted by non-epithelial mammary populations that were underrepresented here (fibroblasts or fat cells) or depend on the synergistic activity of at least two populations.

In conclusion, the current study provides evidence of a novel role for macrophages as CR sensors in the mammary gland that transmit CSF1-mediated signaling for MaSC self-renewal.

## Figures and Tables

**Figure 1 cells-11-02923-f001:**
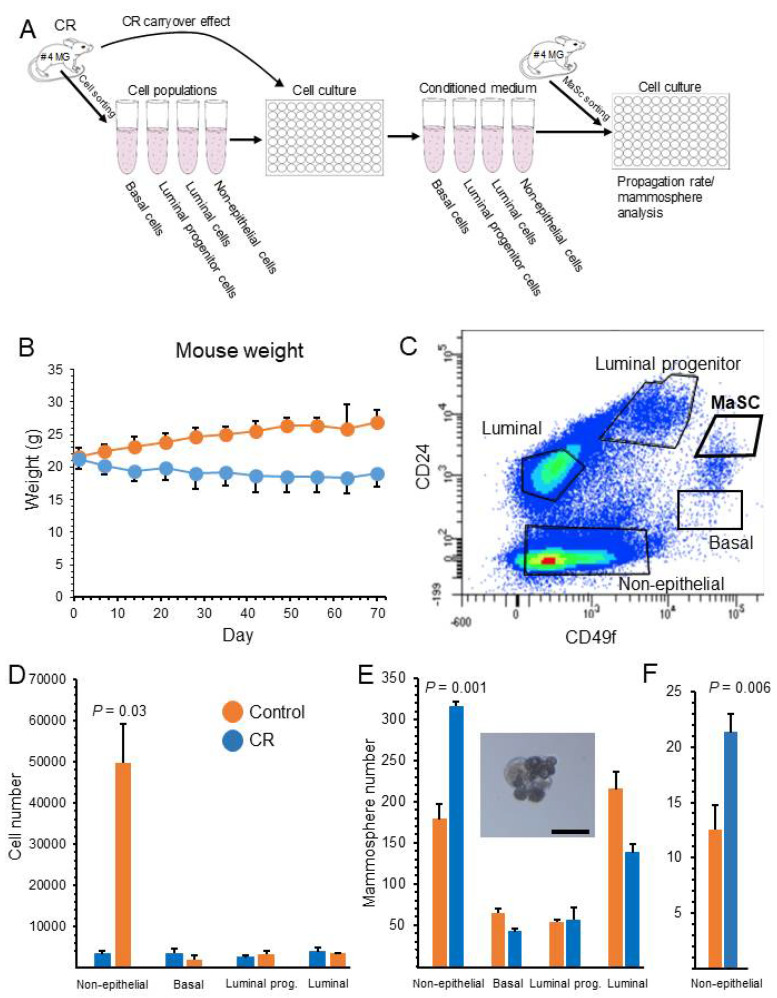
Initial recognition of the non-epithelial cell population as a mediator of the inductive effect of calorie restriction (CR) on mammary epithelial stem cell (MaSC) self-renewal. (**A**) Scheme of the experiment. Four candidate mammary cell populations from ad libitum-fed and CR mice were sorted and cultured for 4 days: basal, luminal, luminal progenitors and non-epithelial. Conditioned medium was collected and incubated with freshly isolated MASCs for 8 days. Cell propagation and mammosphere formation were monitored at later dates. (**B**) A 30% difference in body weight was observed between CR mice and their ad libitum-fed counterparts at the end of the in vivo experimental period. Dots represent mean ± SEM of 20 mice’s body weight. (**C**) Flow cytometry analysis of Lin^-^ mammary cells identified five cell populations. The MaSC population is emphasized. (**D**) Conditioned medium from CR non-epithelial cells induced cell propagation in cultures initiated by MaSCs. Conditioned medium was collected from 12,000 cells/well of each population incubated for 4 days in 100 µL mammary medium in 96-well plates. It was then added to freshly prepared MaSC cultures (100 µL to 500 cells/well). After 12 days of incubation, the cells were counted. Only conditioned medium from CR non-epithelial cells increased cell-propagation rate, compared to its respective control. Bars represent mean ± SEM of 5 replications. Bars represent mean ± SEM of 5 replications. (**E**,**F**) Mammosphere formation was induced after 41 days of incubation (E > 4 cells, F > 10 cells) by conditioned medium of CR non-epithelial cells. See Panel D for the number of seeded cells. Inset: typical mammosphere. Representative demonstration of three independent experiments, starting from the in vivo CR period, is presented. Bars represent mean ± of 5 replications. Bar = 50 µm. MG—mammary gland.

**Figure 2 cells-11-02923-f002:**
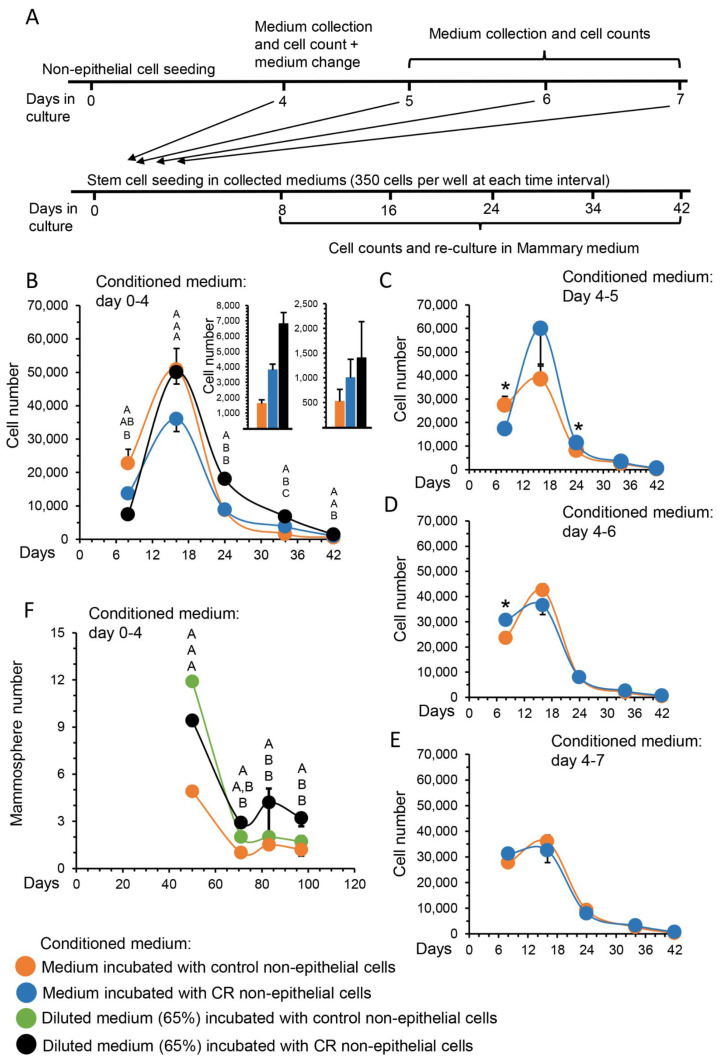
Time-course analysis confirming the involvement of non-epithelial cells in transducing the latent effect of calorie restriction (CR) on mammary epithelial stem cell (MaSC) self-renewal. (**A**) Scheme of the experimental procedure for the propagation rate analysis. Non-epithelial cells were cultured for 4 days in mammary medium (17,000 cells/well in 96-well plate) and conditioned medium was collected as described in Figure 1A. For the first 8 days of culture, 100 µL conditioned medium was supplemented to 350 freshly prepared MaSCs per well. Cells were trypsinized, counted and re-seeded at the equal concentration of 350 cells/well in 100 µL fresh mammary medium to confirm a latent CR effect. This procedure was repeated at each of the indicated times when cell propagation was analyzed. To confirm a time-limited carryover effect, conditioned medium was also collected between days 4 and 7 of the culture period and used to analyze MaSC propagation rate. (**B**) Time-course analysis of propagation rate in MaSC-initiated cultures. Points represents mean ± SEM of cell number, N = 5. Inset: Cell number on days 34 and 42 of culture. Statistical analysis was performed by one-way ANOVA (Tukey–Kramer means comparison). Different letters above the chart indicate statistically significant (*p* < 0.05) differences in the comparison of each value to its counterpart values. (**C**–**E**) Propagation rate was analyzed with conditioned medium collected at later dates from the non-epithelial cell culture. Points represents mean ± SEM of cell number in five replications. A *t*-test was performed to identify statistical differences between treatments. * *p* < 0.05. (**F**) Analysis of mammospheres generated from MaSC-initiated cultures that were supplemented with conditioned medium of non-epithelial cells from ad libitum-fed and CR-treated mice. Points represent mean ± SEM of five replications. Different letters above the chart indicate statistically significant (*p* < 0.05) differences in the comparison of each value to its counterpart values.

**Figure 3 cells-11-02923-f003:**
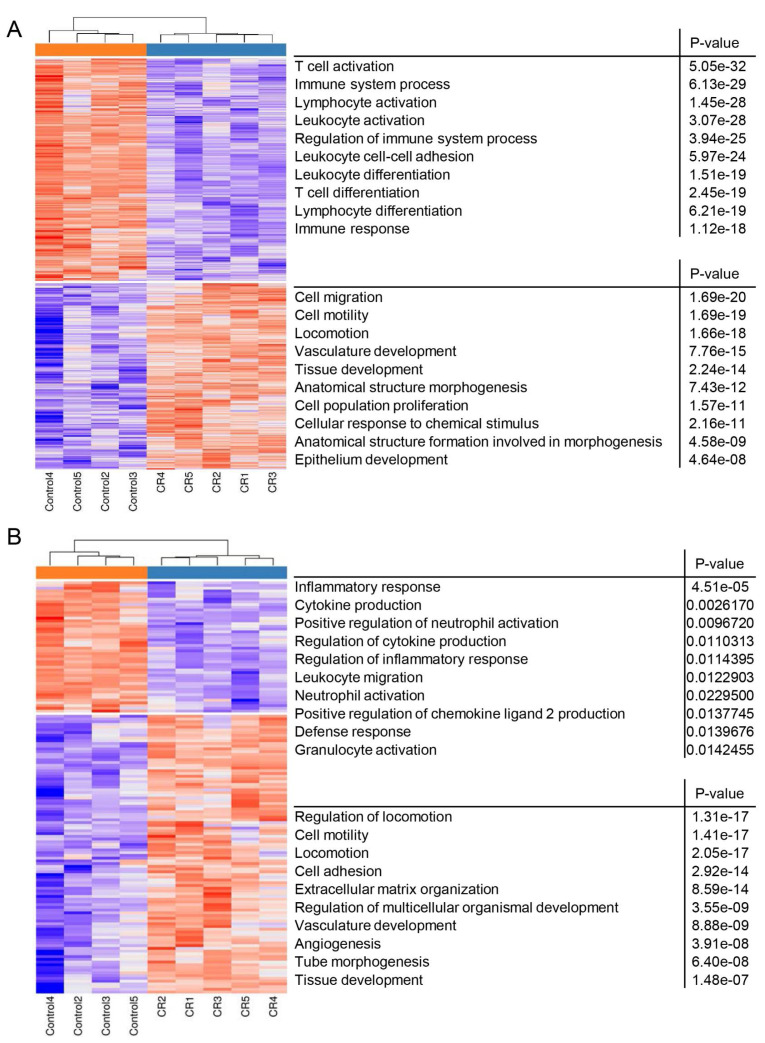
Heatmap and GO analysis presenting enriched biological activities affected by differentially expressed genes in non-epithelial mammary cells derived from ad libitum-fed and calorie restriction (CR)-treated mice. (**A**) Heatmap of all differentially expressed genes. (**B**) Heatmap of differentially expressed genes encoding secreted proteins. Criteria of ≥1.5-fold change, adjusted *p* ≤ 0.05 and minimum 30 counts in at least one of the samples was applied to distinguish the CR effect.

**Figure 4 cells-11-02923-f004:**
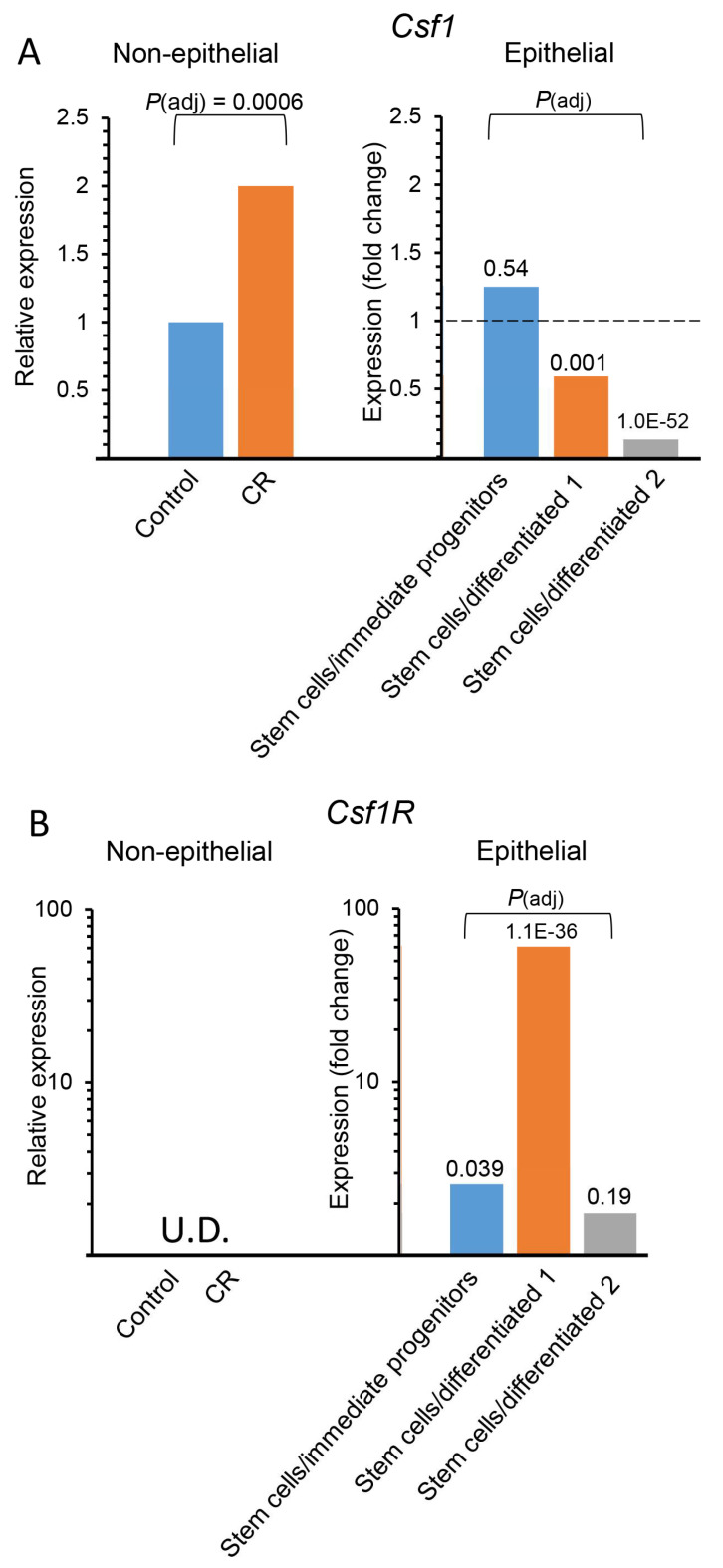
Calorie restriction (CR) induces *Csf1* expression in non-epithelial mammary cells. The analysis of gene expression in the non-epithelial cells was performed in this study. Gene expression analysis in the various stages of epithelial cell differentiation was previously performed [7]. (**A**) *Csf1* expression is induced by CR in the non-epithelial cells relatively to the ad libitum control (control = 1). Its expression is lower in mammary epithelial stem cells (MaSCs), compared to their differentiated cell counterparts. (**B**) *Csf1* receptor *(Csf1R*) expression is barely detectable in non-epithelial cells. It is highly expressed in MaSCs, compared to their common progenitors and differentiated counterparts. Two independent comparisons of *Csf1* or *Csf1R* expression in MaSCs vs. their differentiated epithelial cell counterparts were performed. 1—Stem cells/differentiated cells expressing CD200 but not CD200 receptor (CD200R1). 2—Stem cells/differentiated cells that expressing CD200R1 but not CD200. U.D.—undetected.

**Figure 5 cells-11-02923-f005:**
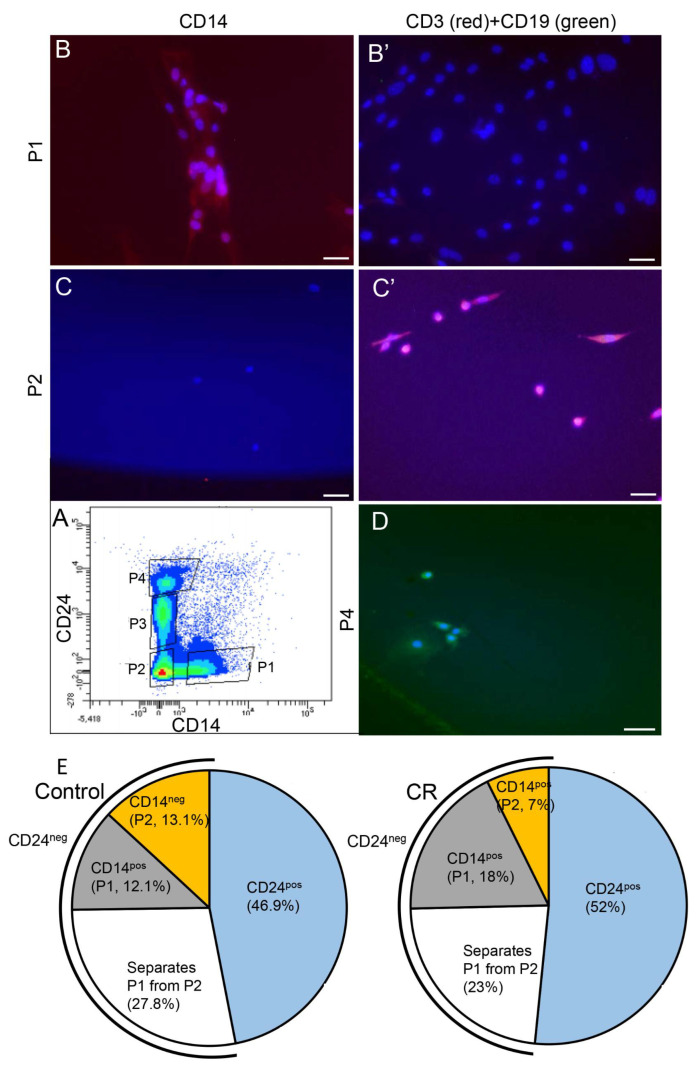
Lin^-^ non-epithelial cell population contains distinguishable myeloid and lymphoid cell subpopulations. (**A**) Flow cytometry analysis separating subpopulations in the non-epithelial cells. (**B**,**B′**) The P1 population (CD14^pos^ CD3^neg^ CD19^neg^) contains mainly monocytes/macrophages, but not lymphoid cells. (**C**,**C′**) The P2 population (CD14^neg^ CD3^pos^ CD19^neg^) contains T cells, but not monocytes/macrophages or B cells. (**D**) The P4 contains B cells. (**E**) Pie charts demonstrating flow cytometry-based analysis of the proportion of each population among the Lin^−^ live mammary cells. Three independent analyses were performed and the mean values are presented.

**Figure 6 cells-11-02923-f006:**
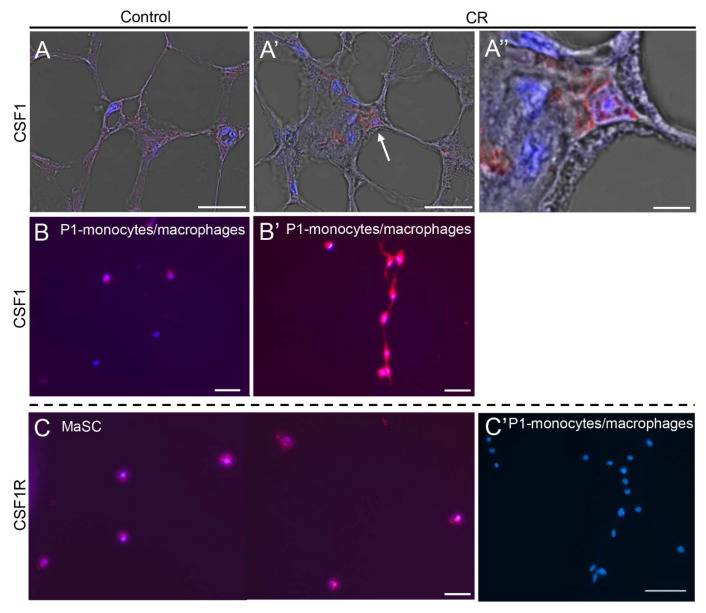
Immunofluorescence analysis demonstrating the inductive effect of calorie restriction (CR) on CSF1 expression in the mammary gland. (**A**–**A″**) CR induces CSF1 expression in the mammary ducts. Arrow indicates CSF1-expressing cell. (**B**,**B′**). CR-induced CSF1 expression in P1 (Figure 4A) monocytes/macrophages cultured for 4 days. (**C,C′**) CSF1R is highly expressed in mammary epithelial stem cells (MaSCs) after attachment but is not detected in monocytes/macrophages of the P1 fraction. Bar = 50 µm.

**Figure 7 cells-11-02923-f007:**
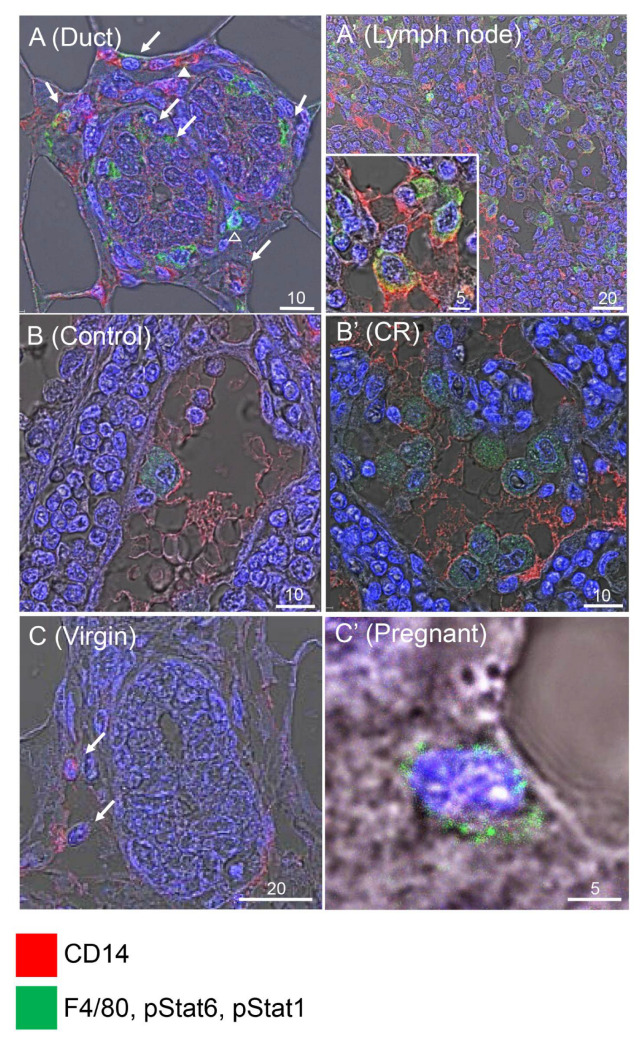
M2 macrophage polarization is induced by calorie restriction (CR). (**A**,**A′**) CD14 and F4/80 are co-expressed in mammary ducts (**A**) and lymph node (**A′**). Arrows indicate cells expressing both CD14 and F4/80. Arrowheads mark rare cells expressing either CD14 or F4/80. Inset, larger magnification of cells co-expressing CD14 and F4/80. (**B**,**B′**) CR induces the number of cells co-expressing CD14 and pStat6. (**C**,**C′**). The pStat1 expression is not detected in the virgin mammary gland by immunofluorescence, but pStat1-expressing cells are detected in the pregnant gland. Representative analysis of five independent mammary glands. Arrows indicate cells expressing CD14. Numbers above bars indicate their range in µM.

**Figure 8 cells-11-02923-f008:**
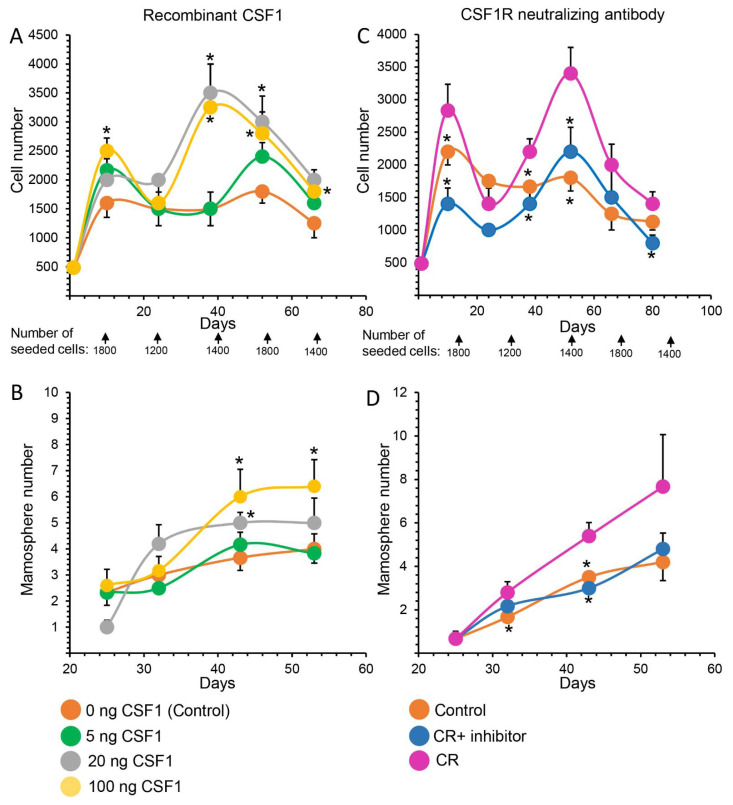
Manipulating the CSF1–CSF1R interaction affects mammary epithelial stem cell (MaSC) self-renewal. (**A**) Latent dose-dependent inductive effect of recombinant CSF1 on cell-propagation rate in cultures initiated by MaSCs. MaSCs (350 cell/well) were seeded in 96-well plates. Recombinant CSF1 was supplemented for the first 8 days in culture (days 1 and 4). Cells were counted and re-seeded in fresh medium at the indicated times. (**B**) Dose-dependent effect of CSF1 on mammosphere formation. Cells were incubated with CSF1 as described in Panel A and mammosphere generation was recorded at the indicated times. (**C**) Time course analysis of the latent effect of CSF1R-neutralizing antibody on cell-propagation rate in cultures initiated by MaSCs. Conditioned medium was collected from CD14^high^CD24^neg^ monocyte/macrophage cultures of 10,000 cells/well containing in 100 µL medium. CSF1R-neutralizing antibody (10 µg/mL) was added for the first 8 days of culture (days 1 and 4). For details see Panel (**A**). (**D**). Dose-dependent effect of CSF1R-neutralizing antibodies (10 µg/mL) on mammosphere formation from MaSCs. For details see Panel (**C**). Dots represent mean ± SEM of 5 replications. * *p* < 0.05.

**Figure 9 cells-11-02923-f009:**
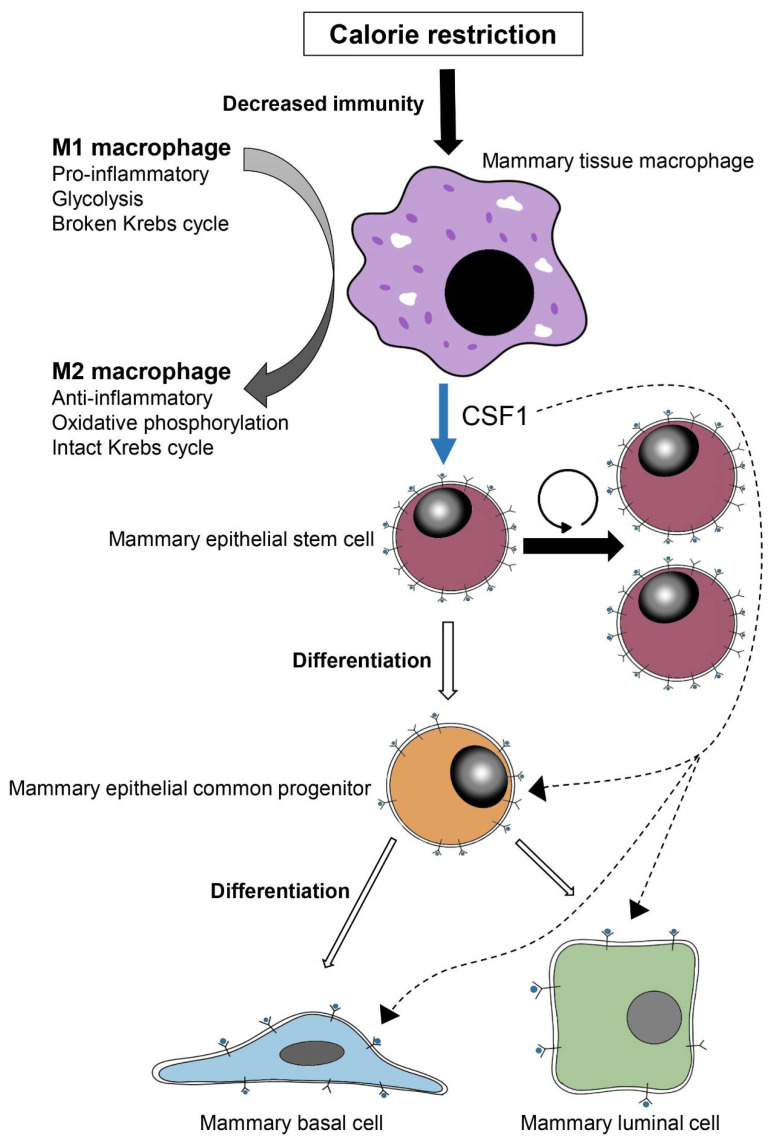
Scheme summarizing study results. The decrease in energy flow to the mammary gland due to calorie restriction (CR) is detected by the residing macrophages and translated into polarization toward the M2 anti-inflammatory and more economic macrophage phenotype. Expression and secretion of CSF1 are then induced. The CSF1 binds CSF1R, which is highly expressed on the surface of the mammary epithelial stem cells (MaSCs), promoting their self-renewal. A distinct effect of CSF1 on mammary epithelial progenitors and their differentiated luminal and basal descendants is probable but was not investigated in this study.

**Table 1 cells-11-02923-t001:** IPA analysis of genes encoding secreted proteins that are highly expressed in calorie-restriction (CR)-treated mice, compared to their ad libitum-fed counterparts and interact with reported stem cell markers.

Analysis	Effector	Secreted Effector	Target Population	Target	Type of Effect	Ref.
#1	Secreted proteins (103 encoding genes)	Csf1	Mouse mammary stem cells (108 genes)	Lgr4, Retnla	Expression	[7]
		Gdnf		Gfra1	Expression	
		Igf1		Pappa, Znf365	Expression	
		Plaur		Itga6	Expression	
#2	Secreted proteins (103 encoding genes)	Csf1	Mouse and human mammary stem cells (43 genes)	Lgr4	Expression	[7]
		Plaur		Itga6	Expression	
#3	Secreted proteins (103 encoding genes)	Brca1	Stem cells from various mouse tissues (36 genes)	Anxa1	Protein–protein interaction	[52]
		Cdca5		Angptl4, Anxa1	Protein–protein interaction	
		Csf1		Tab3	Expression	
		Cxcl12		Dock9	Expression	
		Dcn		BRCA1	Protein–protein interaction	
		Igf1		Brca1, Chek1	Expression Activation	
		Nav2		Emilin2	Protein–protein interaction	
		Rad54B		Igfbp3	Protein–protein interaction	
		Tab3		Emilin1	Protein–protein interaction	
		Trim37		Tnfrsf1A	Protein–protein interaction	
		Trip6		Tle4	Protein–protein interaction	
		Vegfa		Brca1	Protein–protein interaction	

## Data Availability

Data have been deposited in GEO under accession number GEO GSE207481.

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
