# Peer review of "Macrophage-Secreted CSF1 Transmits a Calorie Restriction-Induced Self-Renewal Signal to Mammary Epithelial Stem Cells"

_cells, 2022, doi:10.3390/cells11182923_

Round 1

Reviewer 1 Report

See the file attached

Author Response

Responses to the reviewers’ comments

Editorial request (under a separate cover): Some sections in the Materials and methods section are too similar to previous publications.

Response: We have attempted to make some minor changes in these sections in order to solve this problem.

Common response to both reviewers regarding figures:

During the PDF conversion and manuscript submission process, a decrease in figure quality occurred. We could not inspect this in real time. Therefore, we now provided the editorial office with a separate set of all of the figures in JPEG, under a separate cover, for the reviewers’ inspection. Their quality was very close to that of the original figures which were submitted in TIFF. Please also note that some of the figures combine a bright-field analysis with immunofluorescence. This is essential to orienting cell types within the tissue morphology, but may slightly reduce the sharpness of the figure.

Reviewer #1

 “First of all, the editing is really bad, please check it carefully”.

Response: The manuscript was extensively edited for style and grammar by a highly experienced professional editor before submission and re-edited after revisions. For figures please refer to “Common response to both reviewers” above.

“ Hyperlinks to bibliography do not work”.

Response: The authors do not provide hyperlinks.

 “Materials and methods: in general, this section is poorly described and it is very difficult to follow the experimental scheme”.

Response: We have attempted to provide more details in this section. Please note that for a few of the experimental procedures, we provided references to previous reports of similar types of methodology. Please also note that specific experimental details are provided in the legends to the relevant figures and not in the Materials and Methods section, which mainly includes details that apply to all experiments of the same type.

 “Line 86: how many animals?”

Response: The number of animals is now provided.

“Line 90-91: the references (number / code) of the authorized protocols are missing, please add them, it is not enough to say "in compliance with ..."

Response: This information (including extended details) was transferred to the editorial office under a separate cover. For the convenience of the reviewer and the readers, it is now provided in the text.

“Line 93: #4…do you mean n=4?”

Response. The sign # indicates the number of the mammary gland, i.e., only the number (#) 4 gland was excised from both sides of the female mouse. The female mouse has five glands on each side of its body, which maintain variable properties in terms of milk production. All of the glands, except for #4, are interconnected. To decrease experimental variability, only gland #4 was excised and analyzed.

“Line 94: “Lin cell suspension” … please is necessary an additional explanation”.

Response: The term “Lin-” which appears before the description of the identified cell population is shorthand for “lineage-”. It indicates the application of a commonly used preliminary procedure, prior to cell sorting, that eliminates non-relevant cell populations. We referred to the consequences of this elimination in the Results and Discussion sections and have now added a brief explanation in the Materials and Methods as well.

 “Line 95-105: please re-write this part because it’s absolutely too general. What populations do you want to select? which antibodies did you use? the table S1 is absolutely not sufficient. The information you reported in Figure 1 summarizes the results but the materials and methods must also be clear”.

Response: We have added more details to the sorting description. Because this procedure was applied in various experiments (demonstrated in Figures 1 and 5), the specific details referring to the sorting of individual cell populations are indicated at the relevant places in the Results section and in the legends to these figures.

“Cell culture paragraph. Line 107: cells from each of the sorted populations ... please don't be generic, please clearly identify the groups right away”.

Response: Done

 “Line 116: Haven't you centrifuged the culture medium to eliminate cellular debris before freezing it?”

Response: Cell debris was eliminated. This has been added to the text.

 “Line 121: in most experiments cells were trypsinized…what do you mean in most experiments? Which experiments?”

Response: The sentence has been revised. Cells were trypsinized in all propagation analyses before counting.

 “The paragraph immunohistochemistry and immunofluorescence is very detailed…, why have you neglected the others so much?”

Response: The genomic analysis and bioinformatics are described in detail. We now provide more details and explanations in the sections related to cell sorting and cell culture. Please note that specific details related to individual experiments are provided in the relevant places in the text, and mainly in the figure legends according to the context.

“Results: since the materials and methods are not well described, the results are also confusing, there is not always a correspondence between the groups present in the MM and those shown in the results”.

Response: For the convenience of the reader, a short introductory note appears in each subsection in the Results. Specific experimental details are provided in the relevant section of the Results or in the relevant figure legend, rather than in the Materials and Methods where details of common procedures related to experiments of the same type are reported.

“The quality of figure 1 is very bad! on Figure 1C there is also superimposed something that prevents the graph from being observed”.

Response: The original figures are of high quality. During the conversion to PDF and transfer they lost clarity. Please see the above “Common response to both reviewers”.

“Figure 2: there is also superimposed something that prevents the graph from being observed. Why the medium diluted with CR non-epithelial cells lacks to its appropriate control?”.

Response: Please see the above “Common response to both reviewers”. The appropriate control has now been added to the diluted CR medium.

“The paragraph: 3.3. “CR-Induced CSF1 Expression is Detected in the Mammary M2 Macrophage Population” is a mixed of MM, results and discussion…why?”

Response:  This highly important section defines macrophages as the CR-responsive niche that transfers self-renewal signals to the mammary epithelial stem cells. Six new protein markers and four new cell populations are illustrated and warrant detailed characterization. A summary note has been added for the reader's convenience.

“Figure 4: the same problem of figure 1 and 2 ... same problem as in figure 1 and 2, something is overlapping”.

Response: See “Common response to both reviewers” above.

“Figure 6 and Figure 7 bad quality!”

Response: See “Common response to both reviewers” above.  

 “Discussion: the authors must carefully review the materials, methods and results otherwise the discussion cannot be significative”.

Response: Done

Reviewer 2 Report

The manuscript entitled: "Macrophage-Secreted CSF1 Transmits a Calorie Restriction-Induced Self-Renewal Signal to Mammary Epithelial Stem Cells " focused on the evaluation of how calorie restriction enhances stem cell self-renewal in the mammary gland is well written and requires minor considerations to be accepted for the publication:

- In the introduction section, the authors report the theoretical background behind this application. In my opinion, the authors should improve this section with practical aspects; in particular for breast cancer.

Could the authors analyze the direct relationship between their approach and clinical application? Could the authors explain as calorie restriction was dependent from the other aspects of influence on self-renewal of mammary epithelial stem cells?

In the text, please, could the authors improve for figure 1, 4 and 7?

In the text authors must improve font 

In the Section 2. Materials and Methods.

- 1. Mice and CR

How much mice did use in experiment?

- 2.3. Cell Culture 

The authors must add information about CSF1 in culture.

Author Response

Responses to the reviewers’ comments

Editorial request (under a separate cover): Some sections in the Materials and methods section are too similar to previous publications.

Response: We have attempted to make some minor changes in these sections in order to solve this problem.

Common response to both reviewers regarding figures:

During the PDF conversion and manuscript submission process, a decrease in figure quality occurred. We could not inspect this in real time. Therefore, we now provided the editorial office with a separate set of all of the figures in JPEG, under a separate cover, for the reviewers’ inspection. Their quality was very close to that of the original figures which were submitted in TIFF. Please also note that some of the figures combine a bright-field analysis with immunofluorescence. This is essential to orienting cell types within the tissue morphology, but may slightly reduce the sharpness of the figure.

Reviewer #2

“The manuscript entitled: "Macrophage-Secreted CSF1 Transmits a Calorie Restriction-Induced Self-Renewal Signal to Mammary Epithelial Stem Cells " focused on the evaluation of how calorie restriction enhances stem cell self-renewal in the mammary gland is well written and requires minor considerations to be accepted for the publication”.

Response: We thank the reviewer for this encouraging note.

“In the introduction section, the authors report the theoretical background behind this application. In my opinion, the authors should improve this section with practical aspects; in particular, for breast cancer”.

Response: We already associate CR-induced stem cell self-renewal with longevity in the Introduction. Previous reports have related CR to an anti-tumorigenic effect, but a direct association of the anti-tumorigenic effect of CR with stem cell activity has not been demonstrated. In order keep the focus of the manuscript on relevant aspects of stem cells and their regulatory niche, we limited the text to the following note: “CR has also been recently reported as a non-pharmacological intervention in induced and spontaneous cancers [20]. In a mouse model, it inhibited the growth of induced and spontaneous tumors, including breast cancer [21]. The mechanisms mediating this beneficial effect are not clear; they may involve a unique immune signature of CR and altered activation of metabolic pathways, such as AMPK/mTOR [20,22,23]. Nevertheless, direct association of the anti-tumorigenic effect of CR with stem cell activity has not been demonstrated.”

“Could the authors analyze the direct relationship between their approach and clinical application? Could the authors explain as calorie restriction was dependent from the other aspects of influence on self-renewal of mammary epithelial stem cells?”

Response: CR is not applied in the clinic to treat cancer due to major concerns regarding muscle loss/cachexia. A few clinical experiments are now in progress (Isaac‑Lam and DeMichael, 2022; Meynet and Ricci, 2014; Pomatto-Watson et al., 2021), but these are not directly relevant to the current study. In the Introduction and Discussion, we describe the effect of steroids on mammary epithelial stem cell self-renewal which is transferred from luminal cells and mediated by paracrine secretion of the RANK ligand, Wnt activity and Adamts18 activity. This contrasts with the effect of CR that is mediated by reduced mTOR activity in the intestinal stem cell niche and (here) by CSF1 secretion from macrophages.

“In the text, please, could the authors improve for figure 1, 4 and 7?”

Response: The original figures are of high quality and suffered from the processes of PDF conversion and transfer to the reviewers. See “Common response to both reviewers” above.

 “In the text authors must improve font”.

Response: We are not sure that we understand this comment. Arial 12 is the requested font in many journals.

“In the Section 2. Materials and Methods.

  1. Mice and CR

“How much mice did use in experiment?”
Response: The number of mice is now indicated.

 2.3. Cell Culture 

The authors must add information about CSF1 in culture.

Response: The increasing amounts of CSF1 added to the culture are indicated in Figure 8 (color-coded in the figure as well). The specific procedure of CSF1 supplementation is described in the legend to this figure.

We thank the reviewers for their comments and suggestions.

Round 2

Reviewer 1 Report

The authors soddisfied all my requests.